# A Recurrent Exertional Syncope and Sudden Cardiac Arrest in a Young Athlete with Known Pathogenic p.Arg420Gln Variant in the *RYR2* Gene

**DOI:** 10.3390/diagnostics10070435

**Published:** 2020-06-27

**Authors:** Małgorzata Stępień-Wojno, Joanna Ponińska, Elżbieta K. Biernacka, Bogna Foss-Nieradko, Tomasz Chwyczko, Piotr Syska, Rafał Płoski, Zofia T. Bilińska

**Affiliations:** 1Unit for Screening Studies in Inherited Cardiovascular Diseases, National Institute of Cardiology, 04-628 Warsaw, Poland; mstepien@ikard.pl (M.S.-W.); bfn@ikard.pl (B.F.-N.); 2Department of Medical Biology, National Institute of Cardiology, 04-628 Warsaw, Poland; jponinska@ikard.pl; 3Department of Congenital Heart Diseases, National Institute of Cardiology, 04-628 Warsaw, Poland; kbiernacka@ikard.pl; 4Department of Coronary Artery Disease and Cardiac Rehabilitation, National Institute of Cardiology, 04-628 Warsaw, Poland; tchwyczko@ikard.pl; 52nd Department of Arrhythmia, National Institute of Cardiology, 04-628 Warsaw, Poland; psyska@ikard.pl; 6Department of Medical Genetics, Medical University of Warsaw, 02-106 Warsaw, Poland; rploski@ikard.pl

**Keywords:** sudden cardiac arrest, CPVT, catecholaminergic polymorphic ventricular tachycardia, genetic testing

## Abstract

Catecholaminergic polymorphic ventricular tachycardia (CPVT) is one of causes of sudden cardiac death in the young, especially in athletes. Diagnosis of CPVT may be difficult since all cardiological examinations performed at rest are usually normal, and exercise stress test-induced ventricular tachycardia is not commonly present. The identification of a pathogenic mutation in *RYR2* or *CASQ2* is diagnostic in CPVT. We report on a 20-year-old athlete who survived two sudden cardiac arrests during swimming. Moreover, he suffered repeated syncopal spells on exercise. The diagnosis was made only following genetic testing using a multi-gene panel, and the p.Arg420Gln *RYR2* variant was identified. We present diagnostic and therapeutic issues in this young athlete with CPVT.

## 1. Introduction

Inherited arrhythmia syndromes (e.g., catecholaminergic polymorphic ventricular tachycardia (CPVT)) are common causes of sudden cardiac arrest and cardiac death in the young [1,2,3], especially in athletes [4]. According to ESC guidelines, CPVT is diagnosed clinically in the presence of a structurally normal heart, normal ECG and exercise- or emotion-induced bidirectional or polymorphic VT, or genetically with the identification of pathogenic mutation(s) in the *RYR2* or *CASQ2* genes [1]. The most common type of the disease is related to mutations in the *RYR2* gene, transmitted in the autosomal dominant way, and *CASQ2* mutations account for the recessive form of the disease. CPVT has an estimated prevalence of 1 in 10,000 [1].

We present diagnostic and therapeutic difficulties with regard to a young athlete who suffered repeated syncopal spells and sudden cardiac arrest twice. The patient provided his written informed consent to participate in the study and to publish his data (Bioethics Committee of National Institute of Cardiology, approval no 1407).

## 2. Case Description

A 20-year-old male patient, fit and healthy until the age of 17, with a three-year history of exertional syncope (three times within three years), sought medical attention following a swimming-related syncope, followed by sudden cardiac arrest (SCA). The episode was complicated by aspiration pneumonia, and there were no neurological deficits. Structural heart disease was ruled out with routine noninvasive examinations, including ECG, echocardiography, repeated Holter 24-h ECG monitoring, and coronary angiography. Pharmacologic challenges with flecainide and norepinephrine were negative. The patient had two exercise tests: the first without beta-blocker, in which he attained a maximal load of 10.7 METS (85% of the norm for age and sex) with a maximal heart rate (HR) of 171/min (85% of the maximal HR predicted for age and sex); the second one with beta-blocker, in which the attained load was the same, and his maximal HR was 125/min (63% of the maximal HR predicted for age and sex). During both the exercise tests, there was a progressive appearance of premature ventricular contractions at the mean heart rate threshold of 115 bpm, first isolated and monomorphic, and later in bigeminy, as shown in Figure 1.

Premature ventricular contractions disappeared during the recovery period, and the appearance of arrhythmia was asymptomatic. The patient was advised to take beta-blocker. The family history was not informative. An initial diagnosis of unexplained ventricular fibrillation was made.

Despite suffering from SCA, the patient was unwilling to agree to implantable cardioverter-defibrillator (ICD) implantation and, therefore, an implantable loop recorder was implanted and the patient continued to exercise despite being advised against it. One year later, the patient nearly drowned while swimming, was resuscitated, and the loop recorder tracing showed ventricular fibrillation followed by asystole. With a documented mechanism of sudden cardiac arrest, ICD-DR was implanted for secondary prevention, but had to be removed a year later because of the site’s infection. Within one month, a subcutaneous (S) S-ICD was implanted, and treatment with beta-blocker was advised again (propranolol at the dose of 120mg/day). At this moment, long QT syndrome was suspected, although the patient has never had long QT in 12-lead ECG.

Despite being fully informed of the nature of the disease, the patient tapered off the medication, continued training in basketball and suffered from SCA in the mechanism of ventricular fibrillation, as shown in Figure 2, while on the basketball court. 

Following the adverse event, CPVT was suspected and genetic examination was offered. After obtaining informed consent it was performed with multi-gene panel TruSight Cardio.

A known variant p.Arg420Gln in the N-terminus of *RYR2* gene, as shown in Figure 3, was identified, thus confirming the suspected diagnosis of CPVT.

Following adequate discharge from ICD, the patient agreed to take medication and, with 120 mg of nadolol, he has been free from ICD interventions and syncope for 2 years. 

## 3. Discussion

### 3.1. Difficulties in Diagnosis—The Role of Genetics

Our patient started being symptomatic rather late, at the age of 17 years with increased physical activity, and first experienced SCA at the age of 20 years. The mean age of onset of symptoms in CPVT is 7–12 years, although onset may be as late as 40 years [5], and the majority of patients with CPVT experience syncope or cardiac arrest by their adulthood. Moreover, the diagnostic hallmark of CPVT [1] is exercise- or emotion-induced bidirectional or polymorphic VT, highly specific for CPVT, but this is also observed in Andersen–Tawil syndrome with *KCNJ2* mutations [6], which were not seen in our patient. The drug challenge was inconclusive. Therefore, the diagnosis in our patient was delayed over three years after onset of symptoms. Only genetic examination helped us to make a proper diagnosis.

The replacement of a positively-charged arginine with a polar glutamine at codon 420 of the RYR2 protein identified in our patient, was first reported in two unrelated individuals from a cohort of patients clinically diagnosed with either CPVT or possible long QT syndrome [7]. Subsequently, it was found in the youths aged 7–14 years diagnosed following syncope [8] or cardiac arrest [9]. Similarly as in our patient, Shigemizu et al. identified the p.Arg420Gln variant in a 12-year-old individual with syncope during swimming who later experienced sudden death at 17 years of age [10]. Additionally, this variant has been identified in a large Spanish family, with nine affected relatives with a clinical diagnosis of CPVT who also had a history of sinus bradycardia, atrial and junctional arrhythmias, and/or giant post-effort U-waves [11]. The functional importance of arginine residue at the 420 position is supported by the identification of a pathogenic variant at the same residue (p.Arg420Trp) in association with polymorphic ventricular tachycardia and experimental studies of mice carrying the variant [12]. Furthermore, the p.Arg420Gln variant resides within the N-terminal domain, one of the three hot-spot regions of the *RYR2* gene, where the majority of pathogenic missense variants have been shown to cluster (e.g., p.Thr415Arg, p.Ile419Phe) [7]. Mutations in *RYR2,* responsible for the autosomal dominant form of CPVT, cause a substantial imbalance in the homeostasis of intracellular calcium, resulting in bidirectional or polymorphic ventricular tachycardia through different mechanisms [13]. Increased diastolic SR Ca^2+^ leaks in ventricular myocytes lead to delayed afterdepolarizations and triggered activity via the Na^+^/Ca^2+^ exchanger current, thus promoting ventricular arrhythmia [13]. Another mechanism that seems to be important in triggering arrhythmia is the involvement of Purkinje cells, through increased constitutive intracellular sodium concentration in comparison to ventricular myocytes [13,14].

### 3.2. Therapeutic Issues

In our patient, several therapeutic problems emerged—first of all, poor compliance to lifestyle modifications. Despite two episodes of SCA the patient continued to perform vigorous physical activity. However, in a recent study on competitive sports participation in patients with CPVT, Ostby et al. [15] analyzed outcome data on 63 patients. In total, 31 (49%) of them were athletes at some point of their life. Compared to non-athletes, they were younger at the time of the diagnosis and more symptomatic. However, following the diagnosis, 21 of 24 (88%) athletes continued competition and, during the follow-up, few adverse events were present both in the athletes and non-athletes (p = NS). Luckily, none of the serious adverse events resulted in death [15]. Of note, the earlier a CPVT is diagnosed, the worse the prognosis [5]. 

Second problem with our patient was a poor adherence to advised medical treatment protocols, especially treatment with beta-blocker, which is a cornerstone of the therapy in CPVT [1,16]. When left untreated, the mortality rate in CPVT before the age of 40 is 30% [6]. In one third of patients, cardiac arrest is the first symptom of disease [6]. One of the arguments against using beta-blocker therapy presented by the patient was “I am bradycardiac”. Indeed, CPVT is associated with bradycardia and aberrant sinus node function may also contribute to atrial tachyarrhythmias [16,17]. Flecainide as monotherapy could be an option, but not applied to our patient due to his reluctant attitude to medical treatment. Hayashi et al. [5] analyzed the outcome in 50 probands and 51 relatives with CPVT. The estimated eight-year cardiac event rate was 32% in the total population, and 27% and 58% in the patients with and without beta-blockers, respectively. The absence of beta-blockers and younger age at diagnosis were independent predictors of major adverse events [5]. Additionally, in the study by Ostby et al., following the experience of any serious adverse event during the follow-up, all patients received an adjustment of their medical therapy [15]. Non-adherence to the prescribed medication was found in 60% of serious cardiac events in the study by Miyake et al. [18] Although beta-blockers remain the mainstay of the therapy, there is some evidence that nadolol is superior to other beta-blockers [19,20]. Only after accepting a high dose of nadolol (120mg/day) is the patient free from arrhythmic episodes. Despite pharmacologic treatment, after numerous syncopes and two episodes of SCA, along with ESC guidelines [1], the avoidance of competitive sports was recommended in our CPVT patient. However, Ostby et al. concluded from their study that the risk of sports participation in CPVT patients may be acceptable for a well-treated and well-informed athlete, although 14% of the athletes experience further events [15]. Nevertheless, recreational activities, not associated with any risk of trauma in case of sudden loss of consciousness/ICD discharge, were advised for our patient. 

The third issue that has to be raised relates to ICD therapy in CPVT patients [21]. ICD, in addition to beta-blockers is recommended to patients with a diagnosis of CPVT who experience SCA despite optimal medical therapy [1]. In our patient, ICD was implanted before proper diagnosis and discharged adequately during basketball play, thus saving his life. Since then, he started adhering to medical therapy with no adverse events during the short-term follow-up. Flecainide should be considered in addition to beta-blockers if arrhythmic control in the exercise stress test is incomplete (class IIa) [1]. ESC guidelines also recommend left cardiac sympathetic denervation (LCSD) in CPVT, when recurrent syncope or polymorphic/bidirectional VT/several appropriate ICD shocks occur while patients are on beta-blockers or beta-blockers plus flecainide (class IIb recommendation). There is growing interest in LCSD therapy before ICD implantation, given the increasing evidence of its effectivity [19,22]. Nevertheless, ICDs should be programmed with long delays before discharge since delivered shocks can trigger electrical storms via a vicious circle of adrenergic stimulation in CPVT patients [13,23]. CPVT is characterized by a risk of VF without the need for bradycardia pacing; therefore, when it was possible, S-ICD was implanted in our patient following the removal of the endocardial leads. S-ICD is considered an important option in channelopathies [24,25]. 

Although beta-blockers extend survival in CPVT and bring relief to patients from life-threatening arrhythmia, there are also reports showing that elevating sinus rates with atropine reduces or eliminates exercise-induced ventricular ectopy in patients with CPVT, suggesting that it may be a novel therapeutic strategy in CPVT [26].

## 4. Conclusions

In conclusion, the experience that comes from the history of our patient is that CPVT diagnosis may be challenging, and the p.Arg420Gln variant in the *RYR2* gene is inadvertently related to exercise-induced syncope, in particular, related to swimming and playing basketball. Once our patient started following the recommended beta-blocker therapy and adopted lifestyle modifications, no more syncope or ICD discharges were observed during the two-year follow-up.

## Figures and Tables

**Figure 1 diagnostics-10-00435-f001:**
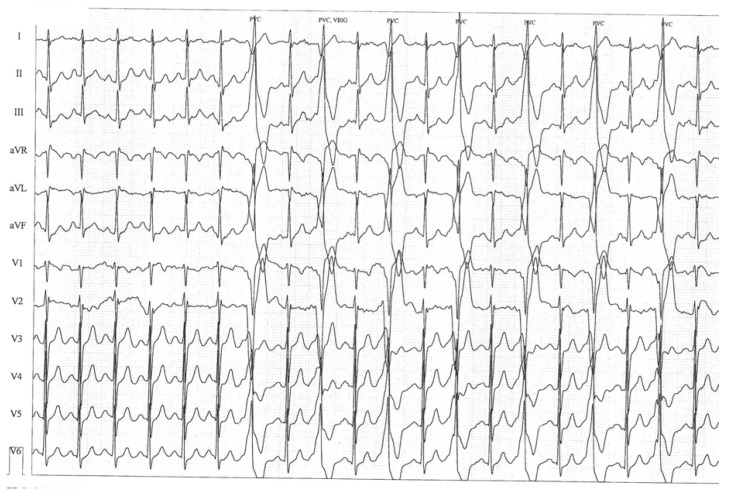
Appearance of ventricular bigeminy during exercise test, which was preceded by isolated premature ventricular contractions.

**Figure 2 diagnostics-10-00435-f002:**
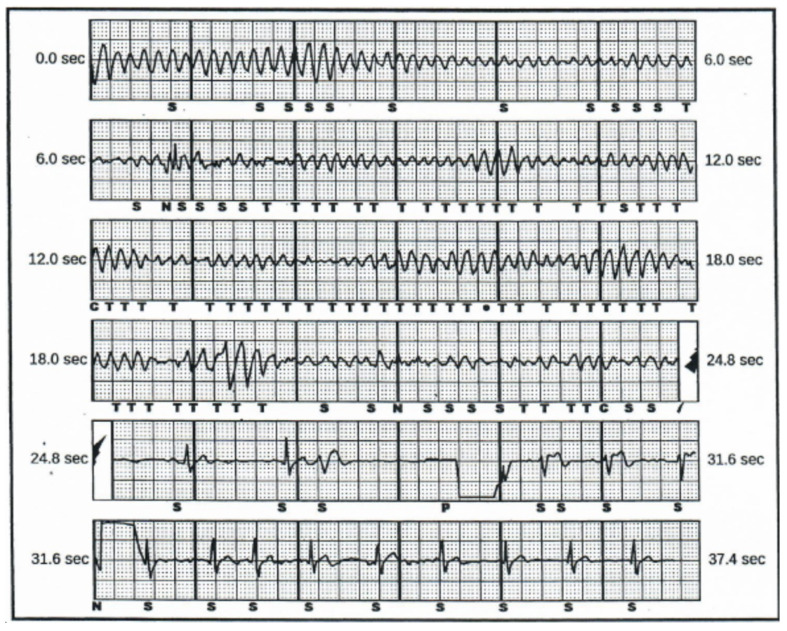
Subcutaneous electrocardiogram showing ventricular fibrillation episode, treated with effective S-ICD shock. Termination of the arrhythmia is followed by sinus beats (S) and, alternately, ventricular extrasystoles of different morphologies (S). One of the beats is a subcutaneously paced beat (P).

**Figure 3 diagnostics-10-00435-f003:**
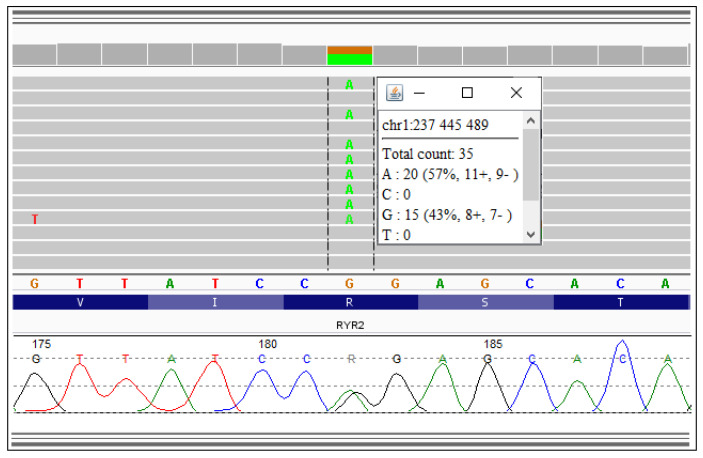
Result of the next generation sequencing and electropherogram showing a variant p.Arg420Gln in the *RYR2* gene.

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
