# Peer review of "A Recurrent Exertional Syncope and Sudden Cardiac Arrest in a Young Athlete with Known Pathogenic p.Arg420Gln Variant in the RYR2 Gene"

_diagnostics, 2020, doi:10.3390/diagnostics10070435_

Round 1
Reviewer 1 Report
Please replace ECHO by echocardiography
Repeated exercice test: how many times was it performed ? did the patient reach the theoretical maximal Heart Rate during theses tests ? what was the maximum power during the tests ?
In the Figure 1, the leads references are missing so we cannot analyse the PVC in terms of axis etc.. please reformat this figure with the corresponding lead before each line
Considering that the patient was supposed to have CPVT after the initial check up, ICD implantation is not recommended in such patients even after SCA, in the absence of previous beta blockers treatment. Please reformulate in accordance to european guidelines on ICD implant in CPVT patients.
Could you explain the initial choice of propranolol which is uncommon is such diseases ?(nadolol usually)
Author Response
Dear Reviewer,
Thank you for review. We answered all your questions, the results are below.
ECHO
Answer: echocardiography
Repeated exercice test: how many times was it performed ? did the patient reach the theoretical maximal Heart Rate during theses tests ? what was the maximum power during the tests ?
Answer: "The patient had two exercise tests, first without beta-blocker, he attained a maximal load of 10.7METS (85% of norm for age and sex) with maximal heart rate (HR) of 171/min (85% of maximal HR predicted for age and sex, and second one on beta-blocker, the attained load was the same, and his maximal HR was 125/min (63% of maximal HR predicted for age and sex). During both exercise tests a progressive appearance of premature ventricular contractions at mean heart rate threshold of 115 bpm, first isolated and monomorphic and later in bigeminy (Figure 1)." (complemented in the manuscript)
In the Figure 1, the leads references are missing so we cannot analyse the PVC in terms of axis etc.. please reformat this figure with the corresponding lead before each line
Answer: Figure 1 was changed
Considering that the patient was supposed to have CPVT after the initial check up, ICD implantation is not recommended in such patients even after SCA, in the absence of previous beta blockers treatment. Please reformulate in accordance to european guidelines on ICD implant in CPVT patients.
Answer: "In our patient ICD was implanted before proper diagnosis, discharged adequately during basketball play, thus saving his life. Since then he started adhering to medical therapy with no adverse events during short-term follow-up."(complemented in the manuscript)
Could you explain the initial choice of propranolol which is uncommon is such diseases ?(nadolol usually)
Answer: "At this moment, the long QT syndrome was suspected, but he has never had long QT in 12-leads ECG." (complemented in the manuscript)
Kind regards,
Małgorzata Stępień-Wojno
Reviewer 2 Report
Authors present the case of a 20 -years old athlete with polymorphic ventricular tachycardia. We considered it of great interest as it is one of the causes of sudden cardiac death during sports activities. The case is very well described ,written and documented with very good figures. Social impact of sudden death in athletes is great what makes it more relevant. Otherwise, this entity should be take into accoun and considered in the diferential diagnosis.
Author Response
Dear Reviewer,
Thank you for the review.
Kind regards,
Małgorzata Stępień-Wojno
Reviewer 3 Report
This is a well-written, nicely documented and informative case of a patient with CPVT. The authors present the case as it occured in real life. In the discussion they comment on the presentation, diagnostics, the role of genetic investigation and therapy by using the relevant evidence from literature. Overall I'm impressed by the manuscript. There are some minor issues in the discussion I highlighted and commented in the manuscript. Please see the attached file. After correcting these issues I could recommend on publishing your interesting case.

Author Response
Dear Reviewer,
Thank you for your helpful remarks. We modified our manuscript accordingly
I suggest to use the term 'ventricular fibrillation'
Answer: fibrillation
The argument that bidir or polymorphic VT is the hallmark of CPVT is true. By citing the review article of Lieve et al, Circ J. 2016 you should specify the following: also bidir VT is highly suggestive of CPVT you can see it also in ATS/KCNJ2. Think CPVT is there are PVC's during CPET: single and in bigeminie as in your patient. In real life things can happen as they did in your patient, but with hindsight you should make things clear. The diagnosis CPVT and genetic diagnostics should have been done following the results of the first exercise test.
Answer: "but also served in Andersen-Tawil Syndrome with KCNJ2 mutations6"
I suggest using the citation of Lieve et al 2016 here too:
When left untreated mortality rate before age 40 is 30%! In 1/3 of patients cardiac arrest is the first symptom of disease.
Answer: We cited the paper by Lieve et al.
"When left untreated mortality rate in CPVT before age 40 is 30%6 . In 1/3 patients cardiac arrest is the first symptom of disease6"
You could discuss here the possibility to switch to flecainide therapy. Its highly effective in combination with BB (this is well documented). Less well documented, but there is some evidence, that its effective as monotherapy too. In a case as yours I would argue that if best prophylaxis isn't accepted by the patient reasonable prophylaxis would be better than none.
Answer: "Flecainide as monotherapy could be an option16 but not applied in our patient due to his reluctant attitude to medical treatment."
Kind regard,
Małgorzata Stępień-Wojno
Round 2
Reviewer 1 Report
The authors have answered all my questions